# CosyCPT: Coreness-Aware Synthetic Continued Pretraining

## Abstract

Synthetic continued pretraining adapts LLMs to specific domains by fine-tuning them on synthetic data that augments real domain data. However, existing methods are often data-inefficient (requiring massive synthetic corpora to enumerate all relational facts) and fail to account for the relative importance of different entity relationships. In this paper, we propose **co**reness-aware **sy**nthetic **c**ontinued **p**re**t**raining (CosyCPT), a systematic pipeline that addresses both limitations. Our method (1) constructs a graph representation of entity relations in a document, (2) quantifies relation importance via coreness scores derived from the graph, and (3) leverages these scores to guide synthetic data sampling and augmentation for continued pretraining. We investigate four definitions of entity coreness and four formulations of relation coreness, verifying that multiple variants of coreness-aware sampling can outperform random sampling of augmented data for synthetic continued pretraining. We offer a mathematical analysis, proving that (1) given a learning budget, maximizing the expected accuracy on a query set about relational knowledge in a document collection is an NP-complete problem, (2) coreness-aware sampling is the optimal solution when each query examines one entity pair, and (3) coreness-aware sampling has a better upper bound for expected accuray than random sampling.

## 1 Introduction

Pre-trained LLMs do not inherently possess knowledge of proprietary or domain-specific corpus. A common remedy is continued pre-training on in-domain corpora in two steps: (i) generate synthetic data to enumerate relational facts contained within the corpus, (ii) fine-tune the LLM on this enriched dataset. Specifically, recent work (e.g., Yang et al., 2024; Zhang et al., 2025) constructs entity graphs from the corpus and generates synthetic data over that graph. While more effective than naive document-level synthesis, these methods remain inefficient: they treat all vertices (entities in graphs) and edges (relation between a pair of entity) equally. In reality, entities and relations contribute unevenly to model downstream performance (e.g., in finance, transaction links between firms may matter more than their headquarters' locations). Ignoring this imbalance wastes synthetic tokens and dilutes training signals. This leads to the central question we aim to study:

*How can synthetic data generation be made more efficient by sampling entities and relations in proportion to their contribution to downstream performance?*

In this work, we introduce coreness-aware synthetic continued pretraining (CosyCPT). Our approach builds on top of the entity graph proposed in Yang et al. (2024) and computes graph-theoretic coreness scores over entities and relations, and uses these scores to guide synthetic data sampling and augmentation. By focusing augmentation on the structural core of domain knowledge, CosyCPT reduces wasted tokens, strengthens training signals, and theoretically requires fewer augmented examples to achieve the same accuracy as uniform enumeration.

Our main contributions are the following:

- We provide a mathematical formulation of the document knowledge acquisition problem, taking into account the fact that entity relations have varying importance in documents.

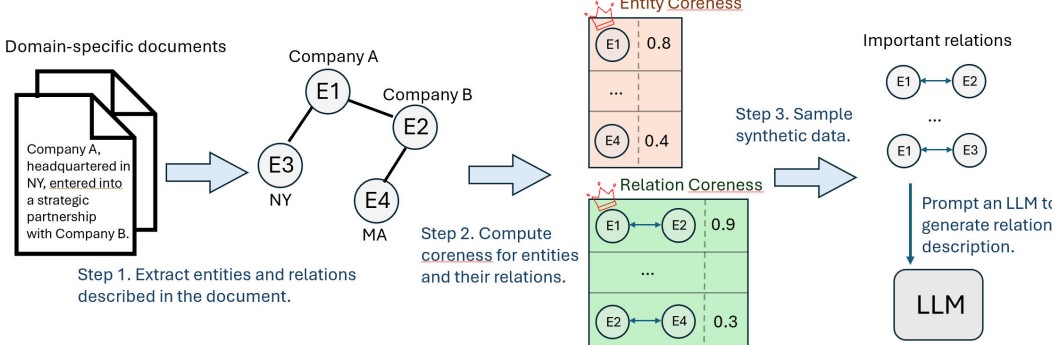

Figure 1: Overview of coreness-aware synthetic continued pretraining (COSYCPT).

- We propose coreness-aware sampling for synthetic continued pretraining, a graph-theoretic framework for modeling entity relation coreness and improve data efficiency of synthetic continued pretraining.
- We empirically demonstrate the effectiveness of coreness-aware sampling, showing up to 4% accuracy improvement on the QuALITY document question-answering task.
- We provide mathematical analyses, proving (1) NP-completeness of maximizing expected accuracy for the problem, (2) optimality of coreness-aware sampling under common assumptions, and (3) a better theoretical bound on expected accuracy compared to random sampling in existing literature.

## 2 RELATED WORK

**Synthetic data.** Progress in LLMs has largely been driven by scaling compute: training bigger models on larger datasets. However, the availability of high-quality data is emerging as a fundamental bottleneck, since the rate at which models learn from data far outpaces the rate at which new data is generated. As a result, curating better training data has become increasingly critical. Synthetic data offer a promising solution and recent studies have shown that these data can effectively supplement real-world corpora to boost model performance. For example, Li et al. (2023); Gunasekar et al. (2023) demonstrate this with Phi-1.5, trained using both web data and synthetic textbooks and exercises data. Another direction focuses on generating synthetic instruction-response pairs, enabling LLMs to better learn how to reliably follow user prompts. For example, self-instruct (Wang et al., 2023) expands a small human seed set by generating and filtering new instruction–response pairs and GLAN (Li et al., 2024) builds synthetic data "from scratch" by structuring human knowledge into subjects and concepts, then producing instructions and exercises with LLMs. This allows for broad coverage beyond existing manually-curated datasets. Beyond generic instruction following, synthetic data has been used to enhance specific skills such as code generation. Work like Magicoder (Wei et al., 2024) produces synthetic instruction data grounded in open-source code snippets; Case2Code (Shao et al., 2025) generates synthetic inductive inference tasks for code (given input-output examples to infer underlying program behavior); UnitCoder builds large, verified code datasets by combining model-generated unit tests with code retrieval.

**Continued pre-training (CPT).** CPT can inject domain-specific knowledge into pre-trained LLMs (Gururangan et al., 2020). Broadly, there are two major directions: (i) continuing pre-training on a diverse set of corpora to enhance general knowledge and reasoning capabilities (Lewkowycz et al., 2022; Chen et al., 2023; Azerbayev et al.; Shao et al., 2024; Colombo et al., 2024; Yuan et al., 2024), and (ii) continuing pre-training on proprietary documents to adapt the model to a targeted domain (Mecklenburg et al., 2024; Yang et al., 2024). In this work, we focus on the latter challenge. The most relevant prior work is by Yang et al. (2024) who construct an entity graph from a small domain corpus, synthesize a larger corpus using this graph, and continue pre-training on this synthetic data. However, they assume that all entities are equally important and uniformly samples across the graph, overlooking differences in relational significance. We improve upon this approach

by introducing a coreness-aware framework that re-weights entities and relations according to their structural importance, enabling more data-efficient sampling for CPT.

**Structuring synthetic data for CPT.** Several works explore how to better control or structure synthetic data for CPT. One line of work investigates the choice of data format. Knowledge-Instruct (Ovadia et al., 2025) proposes using an instruction-following question–answer format to inject new knowledge while mitigating catastrophic forgetting and preserving general reasoning ability. Active Reading (Lin et al., 2025) trains models to "study" domain material through self-generated learning strategies such as paraphrasing, analogy, and active recall. LinkQA (Zhang et al., 2025) represents a general corpus, from mathematics to diverse knowledge domains, through knowledge points. They build a knowledge-point graph from question–answer seeds to synthesize diverse, large-scale datasets, demonstrating improvements in general knowledge and reasoning on MMLU (Hendrycks et al., 2021) and GSM8K (Cobbe et al., 2021) by balancing knowledge coverage and difficulty. While these works primarily focus on designing formats for synthetic data to represent the underlying corpus, we address a complementary problem: efficient knowledge fine-tuning for domain-specific CPT. Our approach quantifies the relative importance of relational facts via coreness scores on entity–relation graphs, thereby prioritizing structurally central knowledge. This effort not only improves efficiency but also enables LLMs to allocate capacity to "core" knowledge rather than uniformly enumerating all relations between entities.

## 3 PROBLEM SETUP

### 3.1 SPECIALIZED KNOWLEDGE ACQUISITION

The primary challenge we address is the efficient adaptation of a pretrained large language model to a domain-specific source document collection, denoted as $\mathcal{D}_{\text{source}}$. The goal is to imbue the model with the specialized knowledge contained in $\mathcal{D}_{\text{source}}$ through continued pretraining. This is typically achieved by first generating a larger, synthetic corpus, $\mathcal{D}_{\text{synth}}$, from the source documents. The effectiveness of the adaptation is then measured by the model's performance on a set of knowledge-intensive test queries, $\mathcal{Q}_{\text{test}}$, which require understanding the information from $\mathcal{D}_{\text{source}}$ without having direct access to it during inference.

### 3.2 A MATHEMATICAL FORMULATION OF KNOWLEDGE ACQUISITION

Our work builds upon the synthetic continued pretraining paradigm established by EntiGraph (Yang et al., 2024). EntiGraph extracts the salient entities $\mathcal{E} = \{enti_1, enti_2, ..., enti_n\}$ from source corpus $\mathcal{D}_{\text{source}}$. Ideally, the relations among entities in any subset $\mathcal{E}' \subseteq \mathcal{E}$ need to be augmented to generate $\mathcal{D}_{\text{synth}}$. With $2^n$ possible subsets of $\mathcal{E}$ and hundreds of salient entities in $\mathcal{D}_{\text{source}}$, it is empirically impossible to enumerate all subsets. To make the problem tractable, EntiGraph focuses on analyzing entity pair relations and presents a log-linear scaling pattern between the number of **uniformly and randomly sampled** entity pairs for augmentation and the knowledge accuracy. We similarly build our mathematical model for analyzing pairwise relations. (**See Appendix A for details**.)

Entigraph's reliance on *uniform random sampling* is the central inefficiency that we aim to address. It implicitly assumes all entity relationships are equally important, leading to wasted computational resources on generating and training on peripheral or redundant knowledge while potentially under-sampling the core concepts of the domain.

Define $\mathcal{I} = \{(enti_i, enti_j)|i, j \in [n], i \neq j\}$. In our mathematical model, we assume there exists a mapping $F$ for each corpus $\mathcal{D}_{\text{source}}$ that assigns each entity pair an importance score

$$F : \mathcal{I} \to [0, 1], \qquad (enti_i, enti_j) \mapsto S_{i,j},$$

defined for all $i, j \in [n]$ with $i \neq j$, such that the scores are normalized: $\sum_{\substack{i,j \in [n] \\ i \neq j}} S_{i,j} = 1$.

To simulate each $S_{i,j}$ ($i, j \in [n]$, $i \neq j$), we first construct a knowledge graph $G = (V, E)$ based on the document $\mathcal{D}_{\text{source}}$. We posit two mappings

$$F_v : \mathcal{E} \to V, \qquad F_v(enti_i) = v_i \quad (i \in [n]),$$

---

**Algorithm 1** COSYCPT: Coreness-Aware Synthetic CPT Pipeline

---

1: **Input:** Source documents $\mathcal{D}_{\text{source}}$, Token budget $B$
2: **Output:** Synthetic corpus for CPT $\mathcal{D}_{\text{synth}}$
3: *// Step 1: Graph Construction*
4: $\mathcal{E} \leftarrow \text{ExtractEntities}(\mathcal{D}_{\text{source}})$
5: $G = (V, E) \leftarrow \text{ConstructGraph}(\mathcal{E}, \mathcal{D}_{\text{source}})$
6: *// Step 2: Coreness Mining*
7: **for** each vertex $v_i \in V$ **do**
8: $\quad Cen(v_i) \leftarrow \text{ComputeEntityCoreness}(G, v_i)$ {See Table 1 & Appx. B}
9: **end for**
10: $\mathcal{S}_{\text{pairs}} \leftarrow \text{Coreness Mining BFS}(G, Cen(\cdot), RC(\cdot, \cdot, \cdot))$ (See Algorithm 2)
11: *// Step 3: Coreness-Aware Sampling*
12: $\mathcal{S}_{\text{ranked}} \leftarrow \text{SortByScore}(\mathcal{S}_{\text{pairs}})$
13: $\mathcal{D}_{\text{synth}} \leftarrow \emptyset$
14: **while** $\text{TokenCount}(\mathcal{D}_{\text{synth}}) < B$ **do**
15: $\quad (v_i, v_j) \leftarrow \text{SampleFromDistribution}(\mathcal{S}_{\text{ranked}})$
16: $\quad \text{text}_{ij} \leftarrow \text{GenerateSyntheticText}(v_i, v_j, \mathcal{D}_{\text{source}})$
17: $\quad \mathcal{D}_{\text{synth}} \leftarrow \mathcal{D}_{\text{synth}} \cup \{\text{text}_{ij}\}$
18: **end while**
19: **return** $\mathcal{D}_{\text{synth}}$

---

$$F_E : \mathcal{R} \to E, \qquad F_E(enti_i, enti_j) = enti_{i,j} \quad ((enti_i, enti_j) \in \mathcal{R}),$$

where $\mathcal{R} \subseteq \mathcal{I}$ denotes the set of entity pairs whose relation is directly stated in $\mathcal{D}_{\text{source}}$.

We denote the distance between $v_i$ and $v_j$ $(i, j \in [n])$ as $Dis(v_i, v_j)$. We define a centrality score $Cen(v_i)$ for each vertex $v_i$ $(i \in [n])$. The centrality score can be simulated by multiple widely applied centrality measures. For details, please see Section 4.3. Hence, to simulate each $S_{i,j}(i, j \in [n], i \neq j)$, we exploit the properties of the knowledge graph structure and propose several heuristic aggregation functions, each incorporating $Cen(v_i), Cen(v_j)$, and $Dis(v_i, v_j)$ as determinants of the score. Detailed discussion will be in Section 4.3.

# 4 COSYCPT: CORENESS-AWARE SYNTHETIC CONTINUED PRETRAINING

To address the data inefficiency of EntiGraph's uniform sampling strategy, we introduce COSYCPT, a pipeline that replaces random sampling with a principled, coreness-aware approach. Our method prioritizes structurally important entities and relations, thereby focusing the synthetic data generation on the core knowledge of the document collection.

## 4.1 METHOD OVERVIEW

The COSYCPT pipeline consists of three main steps, as outlined in Algorithm 1:

1. **Entity-Relation Graph Construction:** We first parse the source documents $\mathcal{D}_{\text{source}}$ to build a knowledge graph $G = (V, E)$, where vertices represent entities and edges represent explicitly stated relationships. We ensure that an edge is added between any two entities only if an explicit relationship exists between the entities in the source documents.

2. **Coreness Mining:** We then analyze the topology of this graph $G$ to simulate a "coreness" score for each entity pair in $\mathcal{I}$.

3. **Coreness-Aware Data Sampling:** Finally, we use these coreness scores to guide the synthetic data generation process. Instead of sampling entity pairs uniformly, we sample them in proportion to their importance, ensuring that the resulting corpus $\mathcal{D}_{\text{synth}}$ consists mostly of necessary core knowledge.

## 4.2 STEP 1: ENTITY-RELATION GRAPH CONSTRUCTION

Following the general approach in Yang et al. (2024), we construct the knowledge graph $G = (V, E)$ through a two-stage process. First, we perform **salient entity extraction** by prompting a large language model (LLM) to identify a comprehensive set of entities, $\mathcal{E} = \{enti_1, enti_2, \ldots, enti_n\}$, from the source documents $\mathcal{D}_{source}$. This set of entities constitutes the graph's vertices $V$.

Subsequently, we perform **explicit edge induction**. For every pair of entities $(enti_i, enti_j) \in \mathcal{I}$, we query an LLM to determine if a direct and explicit relationship between them is asserted in $\mathcal{D}_{source}$. An undirected edge is added to the edge set $E$ if such a relationship exists. The final graph is unweighted, representing the relational structure of the core entities.

## 4.3 STEP 2: CORENESS MINING

Remember the goal is to simulate each $S_{i,j}(i, j \in [n], i \neq j)$ incorporating $Cen(v_i), Cen(v_j)$, and $Dis(v_i, v_j)$ as determinants. Let AG $: \mathbb{R}_{\geq 0} \times \mathbb{R}_{\geq 0} \times \mathbb{N} \to \mathbb{R}_{\geq 0}$ denote the heuristic aggregation function that instantiates $S_{i,j}$. For vertices $v_i, v_j \in V$ with $i \neq j$, we define

$$S_{i,j} = \text{AG}\big(Cen(v_i), Cen(v_j), Dis(v_i, v_j)\big)$$

**Entity Centrality** We first compute a centrality score for each entity (vertex) using well-established graph centrality measures, summarized in Table 1. These metrics leverage the graph's topology to assess a node's structural importance, justified by the observation that knowledge graphs often exhibit topological similarities to social networks, such as sparsity, small-world properties and power-law degree distribution 5. **A detailed discussion of the motivation, definition, and complexity for each measure is provided in Appendix B.**

**Distance** Suppose that we have already computed $Cen(v_i)$ ($\forall i \in [n]$). We enumerate entity pairs by performing breadth-first search (BFS) on the graph $G$. For each source vertex $s \in V$, BFS yields distances $Dis(s, t)$ to all reachable targets $t \in V$. We then record, for each unordered pair $(s, t)$, a 3-tuple $\big(s, t, \text{AG}(Cen(s), Cen(t), Dis(s, t))\big) = \big(s, t, S_{s,t}\big)$. The resulting collection constitutes **entity_pair_info_list** $\mathcal{S}_{pair}$. (**Please see Appendix F for detailed implementation.**)

**Aggregation Functions to Simulate Relation Coreness** We propose several heuristic aggregation functions (See Table 2) to simulate each $S_{i,j}(i, j \in [n], i \neq j)$. Table 2 provides an overview of aggregation functions. We define $Max\_Dis(G) = \max(Dis(v_i, v_j))(i, j \in [n], i \neq j)$ and $Min\_Dis(G) = \min(Dis(v_i, v_j))(i, j \in [n], i \neq j)$. Let $Clo(v_i, v_j) = Max\_Dis(G) - Dis(v_i, v_j) + Min\_Dis(G)(i, j \in [n], i \neq j)$. While $Dis(v_i, v_j)$ indicates how far $v_i$ is away from $v_j$, $Clo(v_i, v_j)$ suggests how close they are on $G$.

Intuitively, the more central $v_i$ and $v_j$ are in $G$, the more the relation $(v_i, v_j)$ will matter. Also, the closer $v_i$ and $v_j$ locate to each other, the more salient the relation $(v_i, v_j)$ will be. Typically, each $S_{i,j}(i, j \in [n], i \neq j)$ should rise with $Cen(v_i), Cen(v_j)$, and $Clo(v_i, v_j)$ and fall with $Dis(v_i, v_j)$.

Since the relative weights among determinants($Cen(v_i)$, $Cen(v_j)$, $Clo(v_i, v_j)$, and $Dis(v_i, v_j)$) remain unknown, we explore multiple heuristic aggregation functions that vary the proportional influence of each factor in Table 2. **The detailed rationale and trade-offs between these functional forms are discussed in Appendix C.**

## 4.4 STEP 3: CORENESS-AWARE DATA SAMPLING AND AUGMENTATION

The computed **entity_pair_info_list** $\mathcal{S}_{pair}$ provides the basis for our guided sampling strategy. We rank all entity pairs in descending order of their scores $\big(s, t, S_{s,t}\big)$. To generate the synthetic corpus, we preferentially sample pairs from the top of this ranked list and prompt an LLM to generate relation descriptions for them, conditioned on the content of $\mathcal{D}_{source}$. This process continues until a predefined token budget is met. By prioritizing high-coreness pairs, we ensure that the model's continued pretraining is focused on the most central and structurally significant relationships in the domain.

Table 1: Overview of entity centralities in our experiments.

| Entity Centrality | Symbol | Formula |
|---|---|---|
| Degree Centrality | $C_{\mathrm{deg}}(v)$ | $C_{\mathrm{deg}}(v) = \dfrac{deg(v)}{n-1}$ ($deg(v)$ is the degree of $v$) (1) |
| Betweenness Centrality | $C_{\mathrm{btw}}(v)$ | $C_{\mathrm{btw}}(v) = \displaystyle\sum_{\substack{s \neq v \neq t \\ s \neq t}} \dfrac{\sigma_{st}(v)}{\sigma_{st}}$, ($\sigma_{st}$ is the number of shortest paths $s \to t$, while $\sigma_{st}(v)$ is the number of shortest paths $s \to v \to t$ (2) |
| Closeness Centrality | $C_{\mathrm{cls}}(v)$ | $C_{\mathrm{cls}}(v) = \dfrac{n-1}{\sum_{u \neq v} \mathrm{Dis}(v,u)}$ (3) |
| PageRank Centrality | $C_{\mathrm{pr}}(v)$ | $C_{\mathrm{pr}}(v) = (1-\alpha)\frac{1}{n} + \alpha \displaystyle\sum_{u \in N^{\mathrm{in}}(v)} \dfrac{C_{\mathrm{pr}}(u)}{\mathrm{deg}^{+}(u)}$ (damping factor $\alpha \in (0,1)$; typically 0.85). (4) |

Table 2: Heuristic aggregation functions for simulation of relation coreness $S_{i,j}$.

| Name | Formula for $S_{i,j}$ |
|---|---|
| Attraction_model_alike | $\dfrac{\mathrm{Cen}(v_i)\,\mathrm{Cen}(v_j)}{\mathrm{Dis}(v_i,v_j)^2}$ (5) |
| Triple_product | $\big(\mathrm{Cen}(v_i)\,\mathrm{Cen}(v_j)\,\mathrm{Clo}(v_i,v_j)\big)^{1/3}$ (6) |
| Harmonic_mean_with_distance | $\dfrac{2}{\mathrm{Dis}(v_i,v_j)\left(\frac{1}{\mathrm{Cen}(v_i)} + \frac{1}{\mathrm{Cen}(v_j)}\right)}$ (7) |
| Max_centrality_over_distance | $\dfrac{\max\{\mathrm{Cen}(v_i),\mathrm{Cen}(v_j)\}}{\mathrm{Dis}(v_i,v_j)}$ (8) |

## 5 EXPERIMENTS

### 5.1 EXPERIMENTAL SETUP

**Data and evaluation**   We evaluate our method on the QuALITY benchmark Pang et al. (2022), using the data split released by Yang et al. (2024).

QuALITY is a long-context multiple-choice QA dataset built from narrative articles and stories drawn from sources such as Project Gutenberg and the Open American National Corpus. Human annotators with advanced degrees in literature or teaching write the questions, which are divided into easy and hard categories depending on the difficulty of annotation.

Our evaluation uses a subsample containing **383,508 tokens**, with documents averaging **4,320 words**. Performance is measured using the **Exact Match (EM)** metric. For consistency, we prompt the model to generate both its reasoning process and the final answer in a fixed format, then parse the answer string to compute exact match accuracy.

We consider two evaluation settings. In the **closed-book setting**, the input query is constructed by prepending the document title to the question, thereby reducing ambiguity, following Yang et al. (2024). In the **open-book setting**, we adopt a retrieval-augmented generation (RAG) approach, retrieving relevant text chunks and prepending them to the query before inference.

**Implementation Details**   We use **Qwen-3-32B** (Yang et al., 2025) as the teacher model for generating entity pairs and their associated discussions, which serve as the basis for our synthetic training corpus. The full set of prompts used in synthetic data generation is provided in Appendix D.

For all experiments, we fine-tune **Llama-3.1-8B-Instruct** (Grattafiori et al., 2024) as the student model. Training is performed using the HuggingFace `Trainer` framework, following the hyper-parameters reported in Yang et al. (2024).

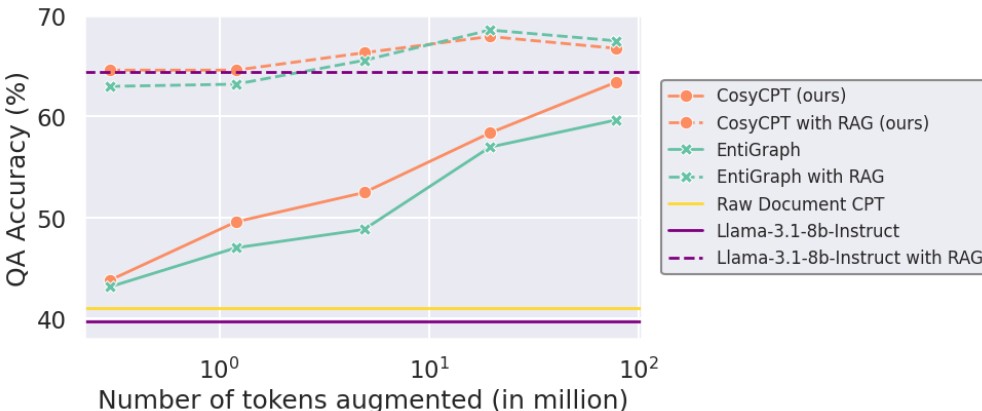

Figure 2: Comparison of our proposed sampling method (CosyCPT with RAG) against EntiGraph random sampling, CPT on raw documents, and prompting with Llama-3.1-8B-Instruct on the QuALITY benchmark. Our method achieves the highest accuracy in the closed-book setting, outperforms Llama-3.1-8B-Instruct in the RAG setting, and performs on par with EntiGraph with RAG.

Evaluation is conducted under both closed-book and open-book (RAG) settings using the publicly released scripts from Yang et al. (2024), ensuring consistency and reproducibility across experimental conditions.

**Baselines**   We compare our method against three baselines: (1) *Prompting the Instruct Model*, where the off-the-shelf Llama-3.1-8B-Instruct is evaluated without additional training to measure its reliance on pretrained knowledge; (2) *Raw Document CPT*, where the model is continued-pretrained for two epochs on the original QuALITY documents; and (3) *EntiGraph CPT*, where continued pretraining is performed for two epochs on a corpus built from uniformly randomly sampled entity pairs augmented with discussions.

## 5.2 MAIN RESULTS

We evaluate our method against three baselines under both **closed-book** and **open-book (RAG)** settings (Figure 2). In the open-book setup, each model is given four retrieved passages, identical across models; the full prompt template is in Appendix E. Baselines include **Raw Document CPT**, which continues pretraining on the unprocessed corpus, and **EntiGraph CPT**, which samples entity pairs with discussion-based augmentation. We also report results for the off-the-shelf **Llama-3.1-8B-Instruct** in both prompting and RAG settings. All methods are trained at multiple data scales ($1\times$, $4\times$, $16\times$, $64\times$, $256\times$).

Across scales, our coreness-based sampling strategy substantially outperforms Raw CPT and EntiGraph, with CosyCPT achieving the highest QA accuracy at $256\times$ tokens. In the open-book setting, both CosyCPT and EntiGraph exceed the naive baselines, which we attribute to document-centric pretraining improving the model's ability to integrate retrieved passages with internalized knowledge. Overall, CosyCPT matches or surpasses EntiGraph while maintaining higher closed-book accuracy, showing that coreness sampling improves both memorization and retrieval-augmented performance.

## 5.3 COMPARING RELATION CORENESS FORMULATIONS

**Setup.**   We compare different strategies for estimating entity coreness and aggregating these scores across entities. For coreness estimation, we ablate over four common graph centrality measures: *degree*, *betweenness*, *closeness*, and *PageRank*. To evaluate aggregation, we consider four alternatives for combining PageRank-based scores: *attraction model alike*, *triple product*, *maximum centrality over distance*, and *harmonic mean with distance*. All results are reported as the change in QA accuracy relative to random sampling.

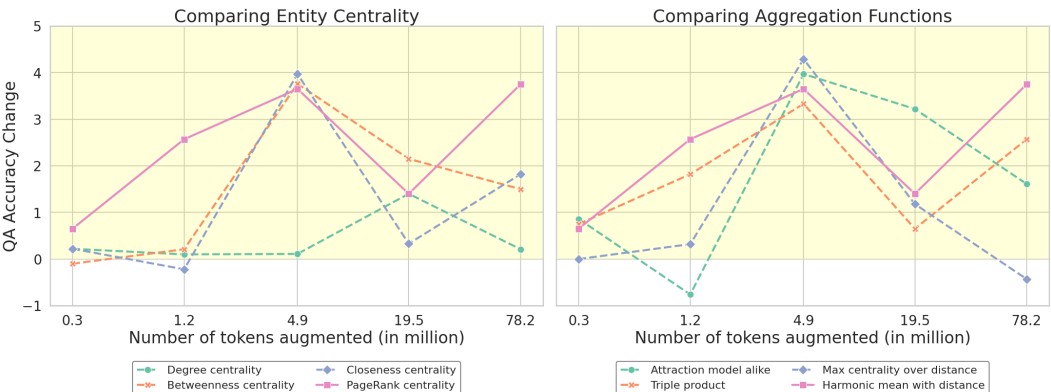

Figure 3: Comparison of entity centrality measures and aggregation functions for coreness sampling. The y-axis reports the change in QA accuracy relative to random sampling. **Left:** Different centrality measures used to identify core entities, where PageRank centrality yields the largest and most consistent improvements. **Right:** Aggregation functions applied to PageRank-based coreness, with the harmonic mean over distance providing the most robust gains.

Table 3: Results across 3 instruction-following benchmarks (scores in %). Overall score is the average across BBH, GPQA, and MMLU-Pro.

| Model | BBH | GPQA | MMLU-Pro | Overall score |
|---|---|---|---|---|
| Llama-3.1-8B-Instruct | 51.01 | 28.77 | 37.78 | 39.19 |
| Naive CPT | 50.86 | 30.05 | 38.52 | 39.81 |
| EntiGraph CPT (Random Sampling) | 49.65 | 28.08 | 36.00 | 37.91 |
| CosyCPT (Ours) | 48.95 | 28.08 | 35.47 | 37.5 |

**Results.** Figure 3 summarizes the results. Among centrality measures, **PageRank** provides the most balanced and consistent improvements across token scales, outperforming degree-, betweenness-, and closeness-based alternatives. With PageRank selected as the best coreness measure, we then compare aggregation strategies. Here, the **harmonic mean with distance** achieves the largest and most stable gains, surpassing both multiplicative and max-based formulations. Together, these findings highlight that PageRank centrality combined with harmonic mean aggregation is the most effective strategy for constructing relation-level coreness.

## 5.4 INSTRUCTION FOLLOWING

**Setup.** We evaluate instruction-following ability using the **lm-eval-harness** implementation Gao et al. (2024) of the Open LLM Leaderboard tasks **Big-Bench Hard (BBH)**, **General Purpose Question Answering (GPQA)**, and **MMLU-Pro**, with results summarized in Table 3. BBH is assessed across all 23 subtasks using the default 3-shot configuration and normalized accuracy (`acc_norm`). GPQA is evaluated in a 0-shot setting with four multiple-choice options, and MMLU-Pro is assessed in a 5-shot setting with ten choices using plain accuracy (`acc`). All evaluations are performed in closed-book mode with no retrieval. For instruction-tuned models, we follow lm-eval recommendations by enabling `--apply_chat_template` and `--fewshot_as_multiturn`. To control for variance, **EntiGraph CPT with Random Sampling** is run with three different seeds, and the reported results are the average across these seeds.

**Results.** Overall, the naive baseline and the Llama-3.1-8B-Instruct baseline perform similarly, while EntiGraph with random sampling shows a modest drop in average score. CosyCPT achieves comparable results, indicating that incorporating coreness-aware sampling retains instruction-following ability, on par with EntiGraph. This complements the open-book findings by showing that CosyCPT maintains stable closed-book performance while introducing a more structured relation sampling strategy.

## 6 THEORETICAL EFFECTIVENESS OF CORENESS-AWARE SAMPLING

Let us now return to the initial setting of our mathematical model. Remember that $\mathcal{I}$ is the collection of entity pairs and $S_{i,j}$ is the importance score for entity pair $(enti_i, enti_j) \in \mathcal{I}$. (See Section 3.2.)

Suppose that the test set contains $T_e$ questions indexed by indexed by $t = 1, 2, \ldots, T_e$. Each question can involve the knowledge of $k$ entities in the context of $\mathcal{D}_{\text{source}}$ ($k \in [n], k \geq 2$ since we assume pairwise entity relation is the minimum knowledge units). Let $q_k$ denote the probability that an item involves exactly $k$ entities. We assume that the entities involved in each test question are sampled with weights derived from pairwise relation importance and increasing with each relation importance scores. More specifically, for any k-set ($k \in [n], k \geq 2$) $\mathcal{S}' = \{enti_{i_1}, enti_{i_2}, \ldots, enti_{i_k}\}$, we can define its combination importance

$$g(\mathcal{S}') = g(enti_{i_1}, enti_{i_2}, \ldots, enti_{i_k})$$

whose score only depends on $\{S_{c,d} | c, d \in \{i_1, i_2, \ldots, i_k\}, c \neq d\}$.

Let $\mathcal{S}_k = \{ \mathcal{S}' \subseteq \mathcal{E} : |\mathcal{S}'| = k \}$ ($k \in [n], k \geq 2$) and define the normalizer

$$Z_k = \sum_{\mathcal{S} \in \mathcal{S}_k} g(\mathcal{S}) \tag{9}$$

Hence, the probability that each question involves exactly the set $\mathcal{S}' \in \mathcal{S}_k$ is

$$\Pr(\text{each question involves exactly } \mathcal{S}') = q_k \cdot \frac{g(S)}{Z_k}, \qquad k \in [n], k \geq 2. \tag{10}$$

Recall that our goal is to maximize the expected accuracy on the test set. For each entity pair $(enti_i, enti_j) \in \mathcal{I}$, Let $W_{i,j} \in \{0, 1\}$ indicate whether the model has learned the corresponding knowledge ($W_{i,j} = 1$) or not ($W_{i,j} = 0$). We write $\binom{\mathcal{S}'}{2} = \{(enti_c, enti_d) : enti_c, enti_d \in \mathcal{S}', c \neq d\}$ for the set of entity pairs in $\mathcal{S}'$. So we have the expected accuracy

$$\mathbb{E}[\text{Acc}] = \frac{1}{T_e} \sum_{t=1}^{T_e} \sum_{k=2}^{n} q_k \sum_{\mathcal{S}' \in \mathcal{S}_k} \frac{g(\mathcal{S}')}{Z_k} \cdot \prod_{(enti_u, enti_v) \in \binom{\mathcal{S}'}{2}} W_{u,v}, \tag{11}$$

We derive three theorems to demonstrate the effectiveness of coreness-aware sampling. For detailed analysis and proof, please see Appendix G.

**Theorem 1 (NP-completeness of maximizing expected accuracy)** *Consider the decision variant of the following problem: given a budget $y$ and a threshold $\tau \in [0, 1]$, does there exist a learned set $\mathcal{L} \subseteq \mathcal{I}$ with $|\mathcal{L}| = y$ such that $\mathbb{E}[\text{Acc} \mid \mathcal{L}] \geq \tau$? This decision problem is NP-complete, even when $q_k > 0$ for a single $k \geq 3$ and $g$ is constant on $\mathcal{S}_k$.*

**Theorem 2 (Optimality of coreness-aware sampling when $q_2 = 1$)** *Suppose $q_2 = 1$ and $q_k = 0$ for all $k \neq 2$. Then for any budget $y$, the* Coreness Sampling *rule that selects the top-$y$ entity pairs by importance score attains an $\mathcal{L}$ that maximizes $\mathbb{E}[\text{Acc} \mid \mathcal{L}]$ among all $\mathcal{L} \subseteq \mathcal{I}$ with $|\mathcal{L}| = y$.*

**Theorem 3 (Coreness-aware sampling has better upper bound than random sampling)** *Let* $\text{UP}(\mathcal{L})$ *be the upper bound on $\mathbb{E}[\text{Acc} \mid \mathcal{L}]$. For any budget $y$ and any value of $q_k$ ($k \geq 2$) and $g(\cdot)$, we have*

$$\text{UP}(\mathcal{L}_{\text{core}}) \geq \text{UP}(\mathcal{L}_{\text{rand}}).$$

## 7 CONCLUSION

We propose coreness-aware synthetic continued pretraining, a graph-theoretic framework to model entity relation coreness in document to guide data sampling and augmentation for document knowledge acquisition. We investigate multiple formulations of entity coreness, aggregation functions, and relation corness, and observe the empirical effectiveness of coreness-aware sampling. We further derive the optimality of our algorithm under common assumptions and its theoretical advantage over random sampling.

## REPRODUCIBILITY STATEMENT

We have taken steps to ensure the reproducibility of our results. We will publish our code used for our experiments upon acceptance. In appendix sections D and E, we provide the prompts we use for reproduciblity purposes. In appendix F we provide algorithmic details to facilitate understanding.

## ETHICS STATEMENT

We are committed to conducting research that adheres to the ethical principles outlined in the ICLR Code of Ethics. Our study focuses on improving the efficiency of existing algorithms and does not cause any intended harm. We are not creating new datasets and comply with relevant data privacy regulations. We have noted below our usage of LLMs in the project.

## LLM CONTRIBUTION

**Entity extraction and relation induction:** During the construction of entity–relation graphs, we prompt an LLM to extract salient entities from source documents and to identify explicit relationships between entity pairs, following EntiGraph (Yang et al., 2024).

**Synthetic data generation:** For coreness-aware augmentation, we use LLM prompting to generate relation-centric text conditioned on pairs of entities and their source document context (see appendix D).

**Assistance:** We leveraged LLM tools for polishing language and for coding assistance such as cursor. All substantive scientific content, experimental design, and analysis were produced by the authors. All generated outputs used in experiments were systematically verified and filtered by the authors before inclusion. The use of LLMs did not replace any of the authors' responsibilities for research design, implementation, or interpretation. We have used ChatBots to discuss some components of our proposed method, such as, the type of possible aggregation functions. But the LLM came up with materials within our knowledge.

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

## A  MATHEMATICAL FORMULATION OF IMPORTANCE SAMPLING

**The Combinatorial Challenge of Full Knowledge Enumeration.**  Our motivation for coreness-aware sampling is grounded in the combinatorial complexity of knowledge enumeration. To capture every possible relational fact within a document, including complex interactions between multiple entities, one would ideally need to generate synthetic data for every possible subset of the $n$ salient entities. This represents a complete but computationally infeasible task, as it requires considering $2^n$ subsets, a number that grows exponentially.

**Simplification to Pairwise Relations: A Massive but Insufficient Step.**  We demonstrate the non-necessity of exhaustively exponential enumeration. Observe that for any $\mathcal{E}_1, \mathcal{E}_2 \subseteq \mathcal{E}$ with $\mathcal{E}_1 \subset \mathcal{E}_2$, the relational knowledge over $\mathcal{E}_1$ is subsumed by that over $\mathcal{S}_2$ and thus need not be augmented repeatedly. A specification of the minimum knowledge for non-redundant enumeration can be defined here to make a significant simplification: assuming that complex knowledge can be sufficiently approximated by the set of all atomic, pairwise facts. This reduces the problem from considering $2^n$ subsets to considering the set of all possible entity pairs, $\mathcal{I} = \{(\text{enti}_i, \text{enti}_j)|i \neq j\}$. While this drastically reduces the complexity from exponential to polynomial ($O(n^2)$), it is still prohibitively expensive to generate synthetic data for every one of the $O(n^2)$ pairs, especially for documents rich in entities.

**Graph Topology as a Principled Proxy for Importance.**  The core issue is that not all pairs are equally important. We hypothesize a "true" but unknown importance distribution over all pairs, represented by a score $S_{i,j}$ $(i, j \in [n], i \neq j)$for each pair. An ideal data generation process would sample pairs according to this distribution. This is the principle of **importance sampling**, and the challenge is to estimate these unknown scores.

We argue that the relational structure of knowledge, as captured by our graph $G = (V, E)$, is a principled proxy for this semantic importance. The structure is not random:

- **Centrality as Influence:** Semantically important entities often serve as hubs or bridges within a knowledge domain. They are more frequently connected to other entities, giving them high centrality.
- **Distance as Relatedness:** Entities that are conceptually close are more likely to have a short path between them in the knowledge graph.

Therefore, we use graph-theoretic properties as a computationally tractable proxy for the latent importance scores. By combining vertex centrality ($Cen(v_i)$) and shortest-path distance ($Dis(v_i, v_j)$), COSYCPT approximates the true importance $S_{i,j}$, enabling a more efficient, guided sampling strategy.$(i, j \in [n], i \neq j)$

## B  DETAILS OF ENTITY CORENESS MEASURES

Here we provide detailed descriptions of the vertex centrality measures summarized in Table 1. Remember that $|V| = n$. We denote $|E| = m$.

### B.1  DEGREE CENTRALITY(FREEMAN (1979))

**Definition**  The degree centrality $C_{\deg}(v)$ is the fraction of nodes a given node $v$ is connected to, calculated as $C_{\deg}(v) = \frac{deg(v)}{n-1}$( equation 1), where $deg(v)$ is the degree of vertex $v$.

**Motivation**  A vertex with higher degree centrality is a local hub, directly referenced by many other entities. While simple, it captures a fundamental notion of influence. However, it may not reflect the global significancy.

**Computation and Complexity**  Computing degree for all vertices requires a single pass through the edge list, leading to a time complexity of $O(n + m)$.

## B.2 BETWEENNESS CENTRALITY(FREEMAN (1977))

**Definition** Betweenness centrality measures the extent to which a vertex lies on the shortest paths between other vertices. It is defined as $C_{\text{btw}}(v) = \sum_{s \neq v \neq t} \frac{\sigma_{st}(v)}{\sigma_{st}}$, where $\sigma_{st}$ is the total number of shortest paths from $s$ to $t$, and $\sigma_{st}(v)$ is the number of those paths that pass through $v$. ( equation 2)

**Motivation** In a knowledge graph, a high-betweenness entity acts as a crucial bridge connecting disparate knowledge clusters. The High-betweenness vertex usually has similar positions with cut vertices or bridges between dense subgraphs. Such entities may not have high degree or a "core" knowledge entity in the conventional sense, but are essential for the flow of information across the graph.

**Computation and Complexity** We compute betweenness centrality using *NetworkX's betweenness_centrality*, which employs Brandes Algorithm (Brandes (2001)). The algorithm iterates over each vertex $s$ ($\forall s \in G$) as a source and runs a BFS to obtain shortest-path distances and all possible predecessors to reachable targets. We then traverse the resulting shortest-path DAG in reverse order to accumulate shortest-path dependencies and update the betweenness scores. This approach avoids the cost of explicitly enumerating $O(n^2)$ pairs of shortest paths. Instead, we do a simple BFS from each source vertex that takes $O(m)$ time, leading to the time complexity of $O(nm)$. For disconnected graphs, vertex pairs in different components are automatically skipped in the path counts.

## B.3 CLOSENESS CENTRALITY(BAVELAS (1950); FREEMAN (1979))

**Definition** Closeness centrality is typically defined as the reciprocal of sum of the distances to every other vertex, often scaled by a factor $n - 1$. Specifically, we have $C_{\text{cls}}(v) = \dfrac{n - 1}{\sum_{u \neq v} \text{Dis}(v, u)}$ ( equation 3).

**Motivation** An entity with high closeness centrality can access all other entities in the graph most efficiently. This indicates it is in a structurally central position within the overall knowledge base. Vertices with high closeness centrality can be a hub knowledge entity of $G$, a pivot of some dense subgraph, or lie on a bridge connecting dense subgraphs slimilar to vertices with high betweenness centrality.

**Computation and Complexity** We compute closeness centrality using *NetworkX*, which conducts a BFS starting from each $v$ ($\forall v \in V$) and sum up the distances to reachable vertice. On disconnected graphs, distances are averaged over the reachable nodes only, and the Wasserman–Faust correction (Wasserman & Faust (1994)) is applied to ensure comparability across components. The total time complexity will be $O(nm)$.

## B.4 PAGERANK CENTRALITY.(PAGE ET AL. (1998))

**Definition** PageRank centrality $C_{\text{pr}}(v)$ ($\forall v \in V$) represents a stationary score of $v$ under a random walk with teleportation. We define $\alpha$ ($0 \leq \alpha leq 1$, typically 0.85) to be a damping factor. The random work follows a random edge at probability $\alpha$ and jumps uniformly to any vertex at probability $1 - \alpha$. Let $N(v)$ be the collection of the neighbors. We have $C_{\text{pr}}(v) = (1 - \alpha)\frac{1}{n} + \alpha \sum_{u \in N(v)} \frac{C_{\text{pr}}(u)}{deg(u)}$.( equation 4)

**Motivation** PageRank centrality aims to capture global significance instead of emphasizing local connectivity. Besides, by introducing teleportation, PageRank Centrality further mitigates the influence of local hub vertex and generates a more globally harmonized score ranking. While $G$ is an undirected graph in our context, PageRank centrality is often applied for directed graphs. However, the teleportation also allows us to handle disconnected graphs more naturally.

**Computation and Complexity** We employ the power iteration method to stimulate the random walk. Specifically, we initialize a $n$-dimensional vector $\vec{x_0}$ with each entry being $\frac{1}{n}$. We define a

matrix $P$ with $P_{i,j}$ being the entry in the i-th row and j-th column of $P$. $P_{i,j} = \frac{\mathbf{1}\{(j,i)\in E\}}{\deg(j)}$. Besides, let $\vec{t_0}$ be a vector with each entry being $\frac{1}{n}$, which is the probability distribution over vertices to which the random work "teleports". We iteratively apply the following formula 12 until the maximum number of iterations $k$ is reached. Let $\vec{x_p}$ be the output vector.

The intuition of equation 12 is to distribute the score of every vertex evenly to its neighbors connected by outgoing edges in each iteration with high probability while with a small chance to evenly distribute the score to every vertex in the graph.

$$\vec{x_{t+1}} = \alpha P \vec{x_t} + (1-\alpha)\vec{t_0} \tag{12}$$

$C_{\mathrm{pr}}(v)$ ($\forall v \in V$) is the $v$-th entry of $\vec{x_p}$. The total time complexity will be $O(kn^2)$ in the worst case. Note that we use a normalized transition probability matrix $P$ instead of adjacency matrix $A$. $\|\vec{x_{t+1}}\|_1 = \|\vec{x_t}\|_1 = 1$ holds for $t \geq 0$. In this case, each entry of $\vec{x_p}$ also represents the probability of staying on the corresponding vertex after the random walk.

## C  RATIONALE FOR AGGREGATION FUNCTIONS

The heuristic aggregation functions in Table 2 explore different ways to combine entity coreness scores and distance, reflecting different assumptions about how these factors interact.

The multiplicative forms in **Attraction_model_alike** and **Triple_product** enforce co-importance; if either entity has a low score, the resulting relation score will also be low. More aggressively, in *Max_centrality_over_distance*, the smaller vertex centrality gets completely dominated by the bigger vertex centrality, and we only care about how extreme one of the centrality values can be. On the contrary, *Harmonic_mean_with_distance* will prevent large factors from dominating the relation coreness score but highlight the impact of small factors, which will improve the robustness under skewed vertex centrality distributions. Compared with *Triple_product*, the **Attraction_model_alike** penalizes distance more heavily with a quadratic term to show more preference for nearby pairs. Additionally, the *Attraction_model_alike* formula is inspired by the classic law of universal gravitation formula.

Note that in the implementation, we normalize $\mathrm{Cen}(\cdot)$ to the scale from $Min\_Dis(G)$ to $Max\_Dis(G)$ to improve numerical stability and enhance comparability between different determinants.

## D  SYNTHETIC DATA GENERATION PROMPTS

**Summary & Entity extraction**  The first step is to summarize and extract the important entities from the document $D$ using the generalized `entity_extraction` operation. The complete `entity_extraction` prompt is as follows:

```
system: As a knowledge analyzer, your task is to dissect and
    understand an article provided by the user.

introduction: |
  You are required to perform the following steps:
  1. Summarize the Article: Provide a concise summary of the entire
     article, capturing the main points and themes.
  2. Extract Entities: Identify and list all significant "nouns" or
     entities mentioned within the article. These entities should
     include but are not limited to:
     * People: Any individuals mentioned in the article, using the
     names or references provided.
     * Places: Both specific locations and abstract spaces relevant to
      the content.
     * Object: Any concrete object that is referenced by the provided
     content.
```

```
      * Concepts: Any significant abstract ideas or themes that are
      central to the article's discussion.

  Try to exhaust as many entities as possible. Your response should
    be structured in a JSON format to organize the information
    effectively. Ensure that the summary is brief yet comprehensive,
    and the list of entities is detailed and accurate.

  Here is the format you should use for your response:

  {
    "summary":  "<A concise summary of the article>",
    "entities": ["entity1", "entity2", ...]
  }

principles: ""

examples: ""

generation: |
  Article:
  {{document}}
```

**Explicit relation existence**  The second step is to analyze and determine if the relation among two or more entities exist explicitly in the document. The prompt we use for generating such a determination of a given entity is as follows:

```
system: |
  You are a helpful assistant that excels at reading comprehension.

introduction: |
  Your task is to classify the existence of an explicitly stated
    relation between two entities in a document. The user will provide
     a document and two entities mentioned in the document. Your role
    is to determine whether the document directly provides any
    information regarding the relation between the two entities.

principles: |
  Respond "Yes" if and only if the document directly discusses an
    interaction, connection, or asssociation between the two entities.
  Respond "No" if the document does not explicitly discuss the two
    entities together, either when there is no relation between them,
    or when their relation is implied, inferred, or requires outside
    knowledge.

  Output your answer in the following JSON format:
  {
    "analysis": "Brief explanation referencing the document",
    "relation": "Yes" or "No"
  }

examples: |
  Here are two examples to help you understand the task:
  ### Example 1:
  Document:
  Batman and the Joker engage ....

  Entity 1: Batman
  Entity 2: Joker

  Question: Is there an explicitly stated relation between Entity 1
    and Entity 2 in the Document?
```

```
Answer:
{
  "analysis": "The document explicitly discusses the conflict and
  psychological battle between Batman and the Joker, highlighting
  their direct interactions.",
  "relation": "Yes"
}

### Example 2:
Document:
Frodo Baggins sets out on a quest ....

Entity 1: Frodo Baggins
Entity 2: Gandalf

Question: Is there an explicitly stated relation between Entity 1
  and Entity 2 in the Document?

Answer:
{
  "analysis": "Although both Frodo and Gandalf are part of the
  broader narrative, the document does not mention any explicit
  interaction between them.",
  "relation": "No"
}

generation: |
  Now it's your turn to perform the task.

  Document:
  {{document}}

  Entity 1: {{entity_pair[0]}}
  Entity 2: {{entity_pair[1]}}

  Question: Is there an explicitly stated relation between Entity 1
    and Entity 2 in the Document?

  Answer:
```

**Coreness-Aware Augmentation** In the final step of the algorithm (Step 3), we use the coreness scores from step 2 to guide the choice of relations that are used for synthetic data generation. Once chosen, we use the following prompt to generate a thorough analysis of the chosen relations:

```
system: |
  You will act as a knowledge analyzer tasked with dissecting a
    document provided by the user.

  Your role involves two main objectives:
  1. Rephrasing Content: The user will identify two specific entities
     mentioned in the document. You are required to rephrase the
    content of the document twice:
      * Once, emphasizing the first entity.
      * Again, emphasizing the second entity.
  2. Analyzing Interactions: Discuss how the two specified entities
    interact within the context of the document.

  Your responses should provide clear segregation between the
    rephrased content and the interaction analysis. Ensure that each
    section of the output includes sufficient context, ideally
```

```
      referencing the document's title to maintain clarity about the
      discussion's focus.

  introduction: ""

  principles: ""

  examples: ""

  generation: |
    Here is the format you should follow for your response:

    ### Discussion of {{document_uid}} in relation to {{entity_pair
      [0]}}
    <Rephrased content focusing on the first entity: {{entity_pair
      [0]}}>

    ### Discussion of {{document_uid}} in relation to {{entity_pair
      [1]}}
    <Rephrased content focusing on the second entity: {{entity_pair
      [1]}}>

    ### Discussion of the interaction between {{entity_pair[0]}} and {{
     entity_pair[1]}} in the context of {{document_uid}}
    <Discussion on how the two entities interact within the document>

    Here are the entities and the document to rephrase and analyze:

    ### Entities:
     - {{entity_pair[0]}}
     - {{entity_pair[1]}}

    ### Document:
    {{document}}
```

## E EVALUATION PROMPTS

Figure 4: Closed-book and open-book input prompt formats for QuALITY.

**Closed-Book Prompt (QuALITY)**

```
According to [ARTICLE TITLE], [QUESTION]
[Options:  (A) ...  (B) ...  (C) ...  (D) ...]
```

**Open-Book Prompt (QuALITY)**

```
[FULL ARTICLE TEXT]
Question:  [QUESTION]
[Options:  (A) ...  (B) ...  (C) ...  (D) ...]
```

## F CORENESS MINING BFS IMPLEMENTATION DETAILS

---

**Algorithm 2** Coreness Mining BFS

---

1: **Input:** Graph $G = (V, E)$, $\{Cen(v)\}_{v \in V}$, $\text{RC}(\cdot, \cdot, \cdot)$
2: **Output:** Entity_pair_shortest_paths_list $\mathcal{S}_{pair}$
3: $\mathcal{S}_{pair} \leftarrow \emptyset$
4: **for** each $s \in V$ **do**
5:    Initialize arrays: $Dis(\cdot) \leftarrow +\infty$; $visited(\cdot) \leftarrow false$
6:    $Dis(s) \leftarrow 0$; $visited(s) \leftarrow true$
7:    Initialize queue $Q = \{s\}$
8:    **while** $Q \neq \emptyset$ **do**
9:      $u \leftarrow Q.popleft()$
10:     **for** each $t$ in neighbors of $u$ **do**
11:       **if** $visited(t) = false$ **then**
12:         $visited(t) \leftarrow true$
13:         $Dis(t) \leftarrow Dis(u) + 1$
14:         $Q.append(t)$
15:         $\mathcal{S}_{pair}.append\big(s, t, \text{RC}(Cen(s), Cen(t), Dis(t))\big)$
16:       **end if**
17:     **end for**
18:    **end while**
19: **end for**
20: **return** $\mathcal{S}_{pair}$

---

## G   Theoretical Effectiveness of Coreness-Aware Sampling

**Analysis Setup and Notation.** Let us now return to the initial setting of our mathematical model. Remember that $\mathcal{I}$ is the collection of entity pairs and $S_{i,j}$ is the importance score for entity pair $(enti_i, enti_j) \in \mathcal{I}$. (See Section 3.2)

Suppose that the test set contains $T_e$ questions indexed by indexed by $t = 1, 2, \ldots, T_e$. Each question can involve the knowledge of $k$ entities in the context of $\mathcal{D}_{source}$ ($k \in [n], k \geq 2$ since we assume pairwise entity relation is the minimum knowledge units). Let $q_k$ denote the probability that an item involves exactly $k$ entities.

**Hypothesis.** Since the model's learning behavior, internal logic, and learning patterns are not directly quantifiable or observable, in Appendix A, we decompose the knowledge of relations among multiple entities into atomic pairwise units. Hence, we postulate the following criterion: *the model answers an question involving entities $enti_{i_1}, enti_{i_2}, \ldots, enti_{i_k}$ ($k \in [n], k \geq 2$) correctly if and only if it has learned all pairwise relation knowledge among $enti_{i_1}, enti_{i_2}, \ldots, enti_{i_k}$.* This also indicates that we assume that (i) there is no partial learning for any pairwise entity relation knowledge, which will also be applied without error once it is learned; (ii) if the model has not learned a specific pairwise knowledge unit, it has no chance to infer from other knowledge or guess the correct answer.

**Test Question Generation via Importance-Weighted Sampling** As we have discussed in Appendix A, test questions are usually biased towards more important knowledge units. To simulate the process of generating test questions based on knowledge importance, we can equivalently assume that the entities involved in each test question are sampled with weights derived from pairwise relation importance and increasing with each relation importance scores. More specifically, for any k-set ($k \in [n], k \geq 2$) $\mathcal{S}' = \{enti_{i_1}, enti_{i_2}, \ldots, enti_{i_k}\}$, we can define its combination importance

$$g(\mathcal{S}') = g(enti_{i_1}, enti_{i_2}, \ldots, enti_{i_k})$$

whose score only depends on

$$\{S_{c,d} | c, d \in \{i_1, i_2, \ldots, i_k\}, c \neq d\}$$

Also, $g$ will have the following monotonicity. For any two sets $\mathcal{S}_1, \mathcal{S}_2 \subseteq \mathcal{E}$ with $|\mathcal{S}_1| = |\mathcal{S}_2|$, suppose there exists a bijection $\phi : \mathcal{S}_1 \rightarrow \mathcal{S}_2$ s.t. for every entity pair $\{enti_c, enti_d\} \subseteq \mathcal{S}_1 (c \neq d)$,

$$S_{c,d} \geq S_{\phi(c), \phi(d)}.$$

Then, we have
$$g(\mathcal{S}_1) \geq g(\mathcal{S}_2).$$

Let $\mathcal{S}_k = \{\, \mathcal{S}' \subseteq \mathcal{E} : |\mathcal{S}'| = k \,\}$ ($k \in [n], k \geq 2$) and define the normalizer
$$Z_k = \sum_{\mathcal{S} \in \mathcal{S}_k} g(\mathcal{S}) \tag{13}$$

Hence, the probability that each question involves exactly the set $\mathcal{S}' \in \mathcal{S}_k$ is
$$\Pr(\text{each question involves exactly } \mathcal{S}') = q_k \cdot \frac{g(S)}{Z_k}, \qquad k \in [n], k \geq 2. \tag{14}$$

**Expected Accuracy** Recall that our goal is to maximize the expected accuracy on the test set. We define a random variable
$$X_t = \mathbf{1}\{\text{question } t \text{ is answered correctly}\} \in \{0,1\}, t \in [T_e]$$

Trivially, we have $\mathbb{E}[X_t] = \Pr(X_t = 1)$. Define the empirical accuracy Acc as
$$\text{Acc} = \frac{1}{T_e} \sum_{t=1}^{T_e} X_t. \tag{15}$$

So we have the expectation
$$\mathbb{E}[\text{Acc}] = \frac{1}{T_e} \mathbb{E}[\sum_{t=1}^{T_e} X_t] = \frac{1}{T_e} \sum_{t=1}^{T_e} \mathbb{E}[X_t] = \frac{1}{T_e} \sum_{t=1}^{T_e} \Pr(X_t = 1). \tag{16}$$

For each entity pair $(enti_i, enti_j) \in \mathcal{I}$, Let $W_{i,j} \in \{0,1\}$ indicate whether the model has learned the corresponding knowledge ($W_{i,j} = 1$) or not ($W_{i,j} = 0$). We write $\binom{\mathcal{S}'}{2} = \{(enti_c, enti_d) : enti_c, enti_d \in \mathcal{S}', c \neq d\}$ for the set of entity pairs in $\mathcal{S}'$. By our hypothesis and importance-weighted sampling principle, we have
$$\Pr(X_t = 1) = \sum_{k=2}^{n} q_k \sum_{\mathcal{S}' \in \mathcal{S}_k} \frac{g(\mathcal{S}')}{Z_k} \cdot \prod_{(enti_u, enti_v) \in \binom{\mathcal{S}'}{2}} W_{u,v}, \tag{17}$$

Consequently, we have expected accuracy
$$\mathbb{E}[\text{Acc}] = \frac{1}{T_e} \sum_{t=1}^{T_e} \sum_{k=2}^{n} q_k \sum_{\mathcal{S}' \in \mathcal{S}_k} \frac{g(\mathcal{S}')}{Z_k} \cdot \prod_{(enti_u, enti_v) \in \binom{\mathcal{S}'}{2}} W_{u,v}, \tag{18}$$

*Remark.* If the questions are i.i.d. under the above mechanism, then $\Pr(X_t = 1)$ is the same for all $t$, and $\mathbb{E}[\text{accuracy}] = \Pr(X = 1)$ for a generic question.

**NP completeness of maximizing expected accuracy**

**Theorem 4 (NP-completeness of maximizing expected accuracy)** *Consider the decision variant of the following problem: given a budget $y$ and a threshold $\tau \in [0,1]$, does there exist a learned set $\mathcal{L} \subseteq \mathcal{I}$ with $|\mathcal{L}| = y$ such that $\mathbb{E}[\text{Acc} \mid \mathcal{L}] \geq \tau$? This decision problem is NP-complete, even when $q_k > 0$ for a single $k \geq 3$ and $g$ is constant on $\mathcal{S}_k$.*

We easily reduce the NP-hard problem $k$-CLIQUE to this problem.

**Theorem 5 (Optimality of coreness-aware sampling when $q_2 = 1$)** *Suppose $q_2 = 1$ and $q_k = 0$ for all $k \neq 2$. Then for any budget $y$, the* Coreness Sampling *rule that selects the top-$y$ entity pairs by importance score attains an $\mathcal{L}$ that maximizes $\mathbb{E}[\text{Acc} \mid \mathcal{L}]$ among all $\mathcal{L} \subseteq \mathcal{I}$ with $|\mathcal{L}| = y$.*

We can easily prove it because of the monotonicity of $g(\cdot)$

**Coreness Sampling vs. Random Sampling**

**Theorem 6 (Coreness-aware sampling has better upper bound than random sampling)** *L    et* $\mathrm{UP}(\mathcal{L})$ *be the upper bound on* $\mathbb{E}[\mathrm{Acc} \mid \mathcal{L}]$. *For any budget $y$ and any value of $q_k$ ($k \geq 2$) and $g(\cdot)$, we have*

$$\mathrm{UP}(\mathcal{L}_{\mathrm{core}}) \;\geq\; \mathrm{UP}(\mathcal{L}_{\mathrm{rand}}).$$

We compare two selection rules under the same budget $y$, i.e., the model will learn $y$ entity pairs from $\mathcal{I}$: (i) *Coreness Sampling*: sort entity pairs by the importance score in a descending order and pick the set $\mathcal{L}_{\mathrm{core}}$ of top-$y$ entity pairs. (ii) *Random Sampling*: pick $\mathcal{L}_{\mathrm{rand}}$ uniformly at random among all $y$-subsets of $\mathcal{I}$ (without replacement).

We define any set of entity pairs $\mathcal{H}$ ($\mathcal{H} \subseteq \mathcal{I}$). Suppose that $\binom{\mathcal{S}_h}{2} = H$ and $|\mathcal{S}_h| = h$. Note that there either exits one unique $\mathcal{S}_h \subseteq \mathcal{E}$ satisfying the above condition, or there does not exist such $\mathcal{S}_h$. If not, we define $w_{\mathcal{H}} = 0$. Otherwise, we define a contribution weight for each $\mathcal{H}$

$$w_{\mathcal{H}} = q_h \cdot \frac{g(\mathcal{S}_h)}{Z_h}$$

For any entity pair set $\mathcal{L} \subseteq \mathcal{I}$, we give the expected accuracy conditioned on $L$

$$\mathbb{E}[\mathrm{Acc}|\mathcal{L}] = \frac{1}{T_e} \sum_{t=1}^{T_e} \sum_{k=2}^{n} q_k \sum_{\mathcal{S}' \in \mathcal{S}_k} \frac{g(\mathcal{S}')}{Z_k} \cdot \prod_{(enti_u, enti_v) \in \binom{\mathcal{S}'}{2}} W_{u,v}$$

$$= \frac{1}{T_e} \sum_{t=1}^{T_e} \sum_{k=2}^{n} q_k \sum_{\mathcal{S}' \in \mathcal{S}_k} \frac{g(\mathcal{S}')}{Z_k} \cdot \mathbf{1}\{ \binom{\mathcal{S}'}{2} \subseteq L \}$$

$$= \frac{1}{T_e} \sum_{t=1}^{T_e} \sum_{\mathcal{H} \subseteq \mathcal{I}} w_{\mathcal{H}} \, \mathbf{1}\{\mathcal{H} \subseteq \mathcal{L}\}$$

In other words, each $\mathcal{H}$ contributes its weight $w_{\mathcal{H}}$ if and only if *all* pairs in $\mathcal{H}$ are present in the learned set $L$; otherwise it contributes zero. Now let us try to distribute the contribution to each entity pair in $\mathcal{H}$. Note that we have

$$\mathbf{1}\{\mathcal{H} \subseteq \mathcal{L}\} \;\leq\; \frac{1}{|\mathcal{H}|} \sum_{(enti_c, enti_d) \in \mathcal{H}} \mathbf{1}\{(enti_c, enti_d) \in \mathcal{L}\}$$

So we have

$$\mathbb{E}[\mathrm{Acc}|\mathcal{L}] \leq \frac{1}{T_e} \sum_{t=1}^{T_e} \sum_{\mathcal{H} \subseteq \mathcal{I}} \frac{w_{\mathcal{H}}}{|\mathcal{H}|} \sum_{(enti_c, enti_d) \in \mathcal{H}} \mathbf{1}\{(enti_c, enti_d) \in \mathcal{L}\}$$

$$= \frac{1}{T_e} \sum_{t=1}^{T_e} \sum_{(enti_c, enti_d) \in \mathcal{L}} \sum_{\substack{\mathcal{H} \subseteq \mathcal{I} \\ (enti_c, enti_d) \in \mathcal{H}}} \frac{w_{\mathcal{H}}}{|\mathcal{H}|}$$

We define $\mu(enti_c, enti_d) = \sum_{\mathcal{H} \ni (enti_c, enti_d)} \frac{w_{\mathcal{H}}}{|\mathcal{H}|}$, $(\forall (enti_c, enti_d) \in \mathcal{I})$

So we have the upper bound $\mathrm{UP}(\mathcal{L})$

$$\mathbb{E}[\mathrm{Acc}|\mathcal{L}] \leq \frac{1}{T_e} \sum_{t=1}^{T_e} \sum_{(enti_c, enti_d) \in \mathcal{L}} \mu(enti_c, enti_d) = \mathrm{UP}(\mathcal{L})$$

Note that $\mu(enti_c, enti_d)$ will increase with $S_{enti_c, enti_d}$ because of the monotonicity of $g(\cdot)$. When $\mathcal{L} = y$, $\mathrm{UP}(\mathcal{L})$ will be maximized if $\mathcal{L} = \mathcal{L}_{core}$. So we will always have

$$\mathrm{UP}(\mathcal{L}_{core}) \geq \mathrm{UP}(\mathcal{L}_{rand}),$$

i.e. coreness sampling method can always have a better upper bound for the expected accuracy.

## H EMPIRICAL FEATURES OF ENTITY RELATION GRAPHS

Figure 5 shows that the entity graphs constructed from real-world documents satisfy social network graph features, such as, low edge density and long-tail connectivity. This motivates the usage of graph-theoretic centrality concepts that are commonly leveraged in social network analysis.

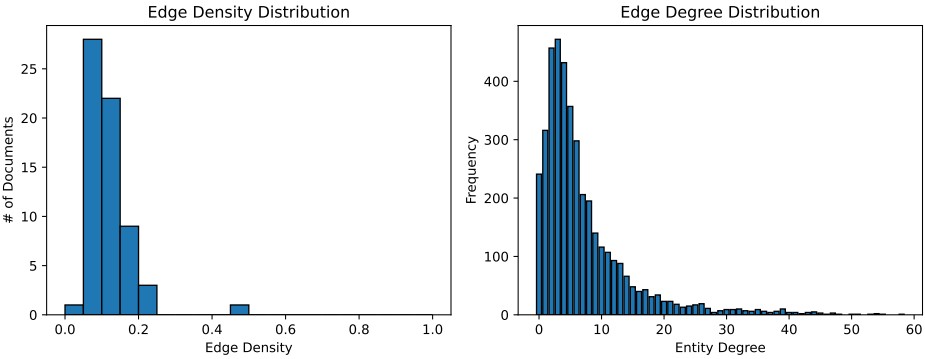

Figure 5: Distributions of edge properties of the entity graph on QuALITY dataset. Left: histogram of edge densities across documents, showing the sparsity of entity graphs. Right: histogram of entity degrees, highlighting the long-tail connectivity pattern where most entities have few links while a small number have disproportionately many.

