# OpenReview forum: "CosyCPT: Coreness-Aware Synthetic Continued Pretraining"
_ICLR.cc/2026/Conference — Submitted to ICLR 2026_

### Official Review · Reviewer_abHG · 2025-10-29

**Soundness:** 3
**Presentation:** 3
**Contribution:** 2
**Rating:** 4
**Confidence:** 4

**Summary:**

The paper introduces COSYCPT, a coreness-aware framework for synthetic continued pretraining (CPT) of large language models. It argues that prior CPT approaches such as EntiGraph sample synthetic relational data uniformly across entities, wasting tokens on unimportant relations. COSYCPT instead models entity importance using graph coreness measures (e.g., PageRank, degree, betweenness, closeness) and samples synthetic data proportionally to these scores.

**Strengths:**

1, The presentation of the paper is good. Math notations and proofs are clear and easy to follow.

2, The motivation that uniform sampling is not efficient makes sense and the proposed method that scores the importance by structual centrality is simple and effective.

3, The experiment results in closed-book setting is convincing, consistently outperforming the baselines.

**Weaknesses:**

1, Limited LLM models and benchmarks in experiments. The paper only uses one model (Llama-3.1-8B-Instruct) and one benchmark (QuALITY benchmark) in the main results, which is not enough to show the priority of the proposed method empirically.

2, The cost of graph construction is not discussed in the paper. The number of LLM APIs / the number of tokens / time cost of graph construction should be added.

**Questions:**

The only question I have is about the limited evaluation (please see weaknesses). I will raise the score if extra experiements on more LLMs or more benchmarks still show the priority over the baseline methods.

---

### Official Review · Reviewer_kiEM · 2025-10-30

**Soundness:** 2
**Presentation:** 2
**Contribution:** 2
**Rating:** 2
**Confidence:** 3

**Summary:**

This paper introduces COSYCPT to improve the data-efficiency of Synthetic Continued Pretraining for LLMs, addressing the inefficiency of prior methods like EntiGraph that rely on uniform random sampling of entity relations . The COSYCPT method first builds an entity-relation graph from source documents . It then uses graph theory, specifically PageRank, to compute a "coreness score" for all entity pairs . Using these scores, it employs coreness-aware weighted sampling to preferentially generate synthetic data about the most important relations . This focused approach is shown to be theoretically optimal under certain assumptions and empirically outperforms random sampling on the QUALITY QA benchmark, particularly in closed-book settings .

**Strengths:**

The paper addresses a practical and significant problem in domain adaptation: the data inefficiency of synthetic continued pretraining . The authors correctly identify a key weakness in prior work, namely that uniform random sampling treats all entity relations as equally important, which can dilute training signals and waste resources . The paper's central premise—that a more efficient path is to selectively sample data based on its structural importance or "coreness" —is intuitive and presents a well-motivated approach to this problem.

**Weaknesses:**

1. Incremental Contribution: The paper's contribution is incremental. The authors merely apply weighted sampling to the existing EntiGraph framework . This is achieved by testing a collection of mature, off-the-shelf graph algorithms (like PageRank ) rather than introducing a novel contribution to the framework itself.

2. Marginal Gain for Sacrificed OOD Performance: The method's claimed improvements appear marginal and come at the cost of OOD generalization, as shown in Table 3. The paper's strategy of focusing on "core" data patterns seems to directly cause this drop in OOD performance—a significant trade-off that is not adequately discussed.

3. Low Figure Quality: The figures are of low quality and detract from the paper's clarity. Figure 1 contains distracting artifacts (e.g., red underlines). Figure 2, which shows the core results, has low resolution and an odd aspect ratio, making scores difficult to read; it should be replaced with a table for clarity. Figure 3 is cluttered and hard to interpret.

**Questions:**

NA

---

### Official Review · Reviewer_aeQX · 2025-10-31

**Soundness:** 1
**Presentation:** 3
**Contribution:** 3
**Rating:** 2
**Confidence:** 4

**Summary:**

The authors consider the problem of compute-efficiency for continued pretraining on synthetic data augmentations. Their approach is to (1) form a knowledge graph over entities contained in documents, (2) apply entity and edge centrality metrics to obtain an importance score for each edge, and (3) to sample synthetic data proportional to these edge importance scores. They compare to random sampling (EntiGraph) on the QuALITY dataset of articles and books, and provide a simple mathematical model justifying their weighted sampling scheme given oracle importance weights.

**Strengths:**

- The paper is generally well-written and clear.
- The problem setting (improving synthetic data generation for continued pretraining) is timely and of interest to the ICLR community.
- The method is intuitive: it is analogous to data curation, focusing pretraining compute on high importance relationships in the text. The task-- and domain-agnostic nature of the graph centrality metrics is also appealing (this could be better articulated by the authors).

**Weaknesses:**

- The authors confusingly refer to their approach as a data-efficiency method. The usual definition of data efficiency in the pretraining and synthetic data literature is in improving performance given a fixed number of unique tokens in a seed corpus [1]. The present work is really an improvement in compute-efficiency, that is, the slope of the "val loss versus log(synthetic tokens)" curve is better. The paper's claims should be rewritten to reflect this; e.g., data-efficiency gains are claimed in Line 3, 70, 109, etc.
- Given the above, the motivation for pursuing compute-efficiency of domain-specific continued pretraining is not clearly presented. For example, the authors could argue that many less well-resourced companies and organizations will want to do synthetic CPT on their proprietary datasets.
- The empirical gains do not appear very significant over random sampling: an inconsistent 1-4% accuracy improvement, on a single dataset (QuaLITY). I would appreciate if the authors could run out larger-scale synthetic CPT runs until one of the approaches begins to asymptote, and ideally add another domain for continued pretraining. This is a particularly useful experiment because if CosyCPT does end up reaching a higher asymptote than EntiGraph, then this paper can actually make a data-efficiency as opposed to compute-efficiency claim (which is more salient in the setting of small proprietary datasets).

[1] Muennighoff et al., 2023. Scaling Data-Constrained Language Models. In NeurIPS.

**Questions:**

Questions
* Can the authors improve the articulation of their main claim, which is improving the compute-efficiency of domain-specific continued pretraining? They should justify why studying compute-efficiency in a data-constrained setting is useful. As a devil's advocate, is it really so many FLOPs to just train on 100x or 1000x the tokens of a small proprietary corpus?
* Alternatively, can the authors run larger-scale experiments which demonstrate CosyCPT reaches a higher asymptotic loss than EntiGraph? This would support the claim that CosyCPT improves data efficiency.
* Can the author run CosyCPT and baselines on another domain (e.g., code or another specialized domain) to support their empirical claims?

I would be willing to increase my score if the improved experiments support the data-efficiency claim or the paper provides a more focused pitch for compute-efficiency.

Clarifications
* EntiGraph constructs entity graphs for each document. It is unclear reading the exposition of your method whether the entity graphs are on a per-document or per-corpus level.
* The 5.4 Instruction Following experiment is just a guardrail to demonstrate that the synthetic CPT does not harm instruction following capabilities? Do you do any replay on instruction tuning data to get this to work?

---

### Official Review · Reviewer_bXXL · 2025-11-01

**Soundness:** 2
**Presentation:** 2
**Contribution:** 2
**Rating:** 4
**Confidence:** 2

**Summary:**

This paper proposes a systematic pipeline, "COSYCPT: Coreness-Aware Synthetic Continued Pretraining". COSYCPT is a systematic graph-theoretic framework designed to prioritize the most structurally important knowledge within a document collection for synthetic data generation.

**Strengths:**

1. This paper provides a clear modular pipeline, and it is relatively easy to reproduce and plug into existing CPT workflows.
2. This paper proves the NP-hardness of the budgeted selection problem. And they show that coreness sampling is optimal when queries involve a single entity pair.

**Weaknesses:**

1. For the assumption mentioned between lines 776 to 779, this paper's method is based on this assumption ("centrality"), and they claimed it is a reliable substitute for its actual semantic importance. However, this may not hold true in all domains; for example, in legal or medical fields [1], a single rarely connected fact could be critically important despite having low centrality.
2. Optimality proof (Line 471 Theorem 2) assumes all queries are pairwise $q_2 = 1$, which is a strong assumption; the relevance of this assumption to real tasks with multi-entity reasoning is unclear. This may oversimplification of knowledge.
3. In the experiment section, this paper only includes a single dataset QUALITY, and a single task QA (English). This is relatively hard to claim that this method has a strong performance that will generalize to other types of documents, such as other languages beyond English.

[1] He X, Zhang J. Why do hubs tend to be essential in protein networks?. PLoS genetics. 2006 Jun;2(6):e88.

**Questions:**

Please explain the weaknesses.

---

### Meta-Review · Area_Chair_3rk6 · 2026-01-21

**Summary:**

This paper presents a novel coreset sampling strategy to guide synthetic dataset sampling and augmentation for continued pre-training of large language models. Reviewers found the method to be an incremental contribution; and found the empirical improvement of the method over a random sampling baseline to be marginal and weakly validated, as the experimental validation was limited to a single model and benchmark in one domain. These concerns were not addressed in a rebuttal.

**Reviewer Concerns:**

All concerns are open:
- reasonableness of the theoretical assumptions
- the marginal improvement over random sampling
- the limited scope of the experiments

**Reviewer Scores:**

No rebuttal was given, so no changes are expected.

---

### Decision · Program_Chairs · 2026-01-26

Reject